# Increasing Black, Indigenous and People of Color participation in clinical trials through community engagement and recruitment goal establishment

Michele P. Andrasik[1]*, Gail B. Broder[1], Stephaun E. Wallace[1], Richa Chaturvedi[2], Nelson L. Michael[3], Sally Bock[4], Chris Beyrer[5], Linda Oseso[1], Jasmin Aina[1], Jonathan Lucas[6], David R. Wilson[7], James G. Kublin[1], George A. Mensah[8]

1 Vaccine and Infectious Disease Division, Fred Hutchinson Cancer Research Center, Seattle, WA, United States of America, 2 Statistical Center for HIV/AIDS Research and Prevention, Fred Hutchinson Cancer Research Center, Seattle, WA, United States of America, 3 Walter Reed Army Institute of Research, Silver Spring, MD, United States of America, 4 Fred Hutchinson Cancer Research Center, Seattle, WA, United States of America, 5 John's Hopkins Bloomberg School of Public Health, Baltimore, MD, United States of America, 6 HIV Prevention Trials Network, FHI360, Research Triangle, NC, United States of America, 7 Tribal Health Research Office, National Institutes of Health, Bethesda, MD, United States of America, 8 National Heart, Lung, and Blood Institute, National Institutes of Health, Bethesda, MD, United States of America

☯ These authors contributed equally to this work.
‡ These authors also contributed equally to this work.
* mandrasik@fredhutch.org

**Data Availability Statement:** All relevant data are within the paper and held in a public repository found at https://atlas.scharp.org/cpas/project/

## Abstract

Longstanding social and economic inequities elevate health risks and vulnerabilities for Black, Indigenous and People of Color (BIPOC) communities. Engagement of BIPOC communities in infectious disease research is a critical component in efforts to increase vaccine confidence, acceptability, and uptake of future approved products. Recent data highlight the relative absence of BIPOC communities in vaccine clinical trials. Intentional and effective community engagement methods are needed to improve BIPOC inclusion. We describe the methods utilized for the successful enrollment of BIPOC participants in the U.S. Government-(USG)-funded COVID-19 Prevention Network (CoVPN)-sponsored vaccine efficacy trials and analyze the demographic and enrollment data across the efficacy trials to inform future efforts to ensure inclusive participation. Across the four USG-funded COVID-19 vaccine clinical trials for which data are available, 47% of participants enrolled at CoVPN sites in the US were BIPOC. White enrollment outpaced enrollment of BIPOC participants throughout the accrual period, requiring the implementation of strategies to increase diverse and inclusive enrollment. Trials opening later benefitted considerably from strengthened community engagement efforts, and greater and more diverse volunteer registry records. Despite robust fiscal resources and a longstanding collaborative and collective effort, enrollment of White persons outpaced that of BIPOC communities. With appropriate resources, commitment and community engagement expertise, the equitable enrollment of BIPOC individuals can be achieved. To ensure this goal, intentional efforts are needed, including an

HVTN%20Public%20Data/CoVPN/CoVPN%20CE
%20Data/begin.view?

**Funding:** All authors were funded by the NIH
Institutes of Health through the COVID-19
Prevention Network. Dr. Chris Beyrer received
support from the CoVPN for his community
engagement work.

**Competing interests:** The authors have declared
that no competing interests exist.

emphasis on diversity of enrollment in clinical trials, establishment of enrollment goals, ongoing robust community engagement, conducting population-specific trials, and research to inform best practices.

## Introduction

The persistent and pervasive health inequities experienced by Black, Indigenous and People of Color (BIPOC) communities are well documented [1, 2]. Longstanding structural inequities elevate health risks and vulnerabilities [3]. When faced with infectious diseases, disparities in morbidity and mortality rapidly emerge in BIPOC communities. A study examining US-based vaccine trials registered on ClinicalTrials.gov from July 1, 2011 through June 30, 2020 found that, among adult studies, Black, African American, American Indian, Alaska Native and Hispanic/Latino/a individuals were underrepresented compared with US census data [4]. In pediatric trials, Black, African American and Hispanic/Latino/a participants were underrepresented. In addition, among the pediatric trials reporting race and ethnicity, almost half did not report American Indian or Alaska Native participants and over 60% did not include Hawaiian or Pacific Islander participants [4]. Effectively engaging BIPOC communities in clinical research is critical to addressing the history of ethical research abuses and the development of trustworthy reputations and relationships. Establishing these conditions should increase vaccine confidence, acceptability, and uptake when approved products become available, thereby strengthening public health.

When the HIV Vaccine Trials Network (HVTN) Leadership Operations Center became part of the COVID-19 Prevention Network (CoVPN) in March 2020, the Community Engagement Team led community engagement efforts for the US Government (USG)-funded COVID-19 vaccine efficacy trials. Longstanding HIV community engagement efforts enabled a quick pivot to address COVID-19.

A robust community engagement effort necessitates relationship building, trustworthiness, and bidirectional communication. The HVTN has worked to center community engagement across its preventive HIV vaccine trial efforts since its founding in 1999. Community members are at the heart of these efforts; and without community, moving impactful science forward is impossible. Central to these efforts is the utilization of Good Participatory Practice (GPP) [5] as a framework and behavioral theories [6–8] to guide the work. In 2011, the Joint United Nations Programme on HIV/AIDS (UNAIDS) and AVAC developed the GPP guidelines to standardize practices globally for stakeholder engagement in biomedical HIV prevention trials. "The GPP guidelines set global standard practices for stakeholder engagement. When applied during the entire lifecycle of a biomedical trial, they enhance both the quality and outcomes of research [5]," The GPP framework has recently been adapted by the World Health Organization to emerging pathogens [9], Across the HVTN, particularly at clinical research sites, community engagement is a collective responsibility shared by persons in all roles—investigators, community staff, clinicians, and Community Advisory Board (CAB) members—and across the entire lifecycle of a research endeavor. CAB members are also included on every HVTN protocol team, operational and scientific committee, and working group. Support for building community engagement capacity is available to CAB members to ensure meaningful engagement and contributions.

All clinical research sites are required to have active community advisory groups with clear lines of communication to clinical research site staff and leadership. Clinical research sites are also required to develop annual work plans that outline processes and goals for community

engagement efforts with measurable objectives that are reviewed and approved by the HVTN Community Engagement Team. Community Working Groups (CWGs) comprised primarily of community staff and CAB members are convened to provide guidance and direction for all efficacy trials.

Additionally, the HVTN conducts ongoing mixed methods studies to inform an increased understanding of barriers and facilitators to research participation for populations most impacted by HIV [10–16]. In 2017, we examined demographic characteristics across Phase 1/2a preventive HIV vaccine studies conducted in the US [17]. Prioritizing community partnerships and investing resources in community engagement showed a 94% increase in enrolled participants who identified as a member of a racial/ethnic minority group. This increased from 17% in trials conducted between 1988 and 2002 [18] to 32.7% in trials conducted between 2002 and 2016.

Recent data illustrate the need for effective efforts to ensure equitable inclusion of BIPOC communities in vaccine clinical trials [4]. BIPOC communities are disproportionately impacted by COVID-19 cases, hospitalizations, and death [19, 20]. Ensuring BIPOC enrollment in the COVID-19 vaccine trials was critical to ensure that the vaccines were evaluated in the context of their intended use, and to support inclusive vaccine acceptance and uptake efforts. We describe the methods utilized for the successful enrollment of BIPOC participants in the US government-funded COVID-19 vaccine efficacy trials and analyze the related demographic and enrollment data to inform future efforts on inclusive participation.

## Materials and methods

Pivoting to COVID-19 work required reaching out to new and existing partners, engaging in conversations to understand how COVID-19 was impacting their respective communities, exploring barriers to trial participation and challenges to vaccine confidence and acceptability, and identifying processes to ensure community input into research protocols and community engagement efforts. These conversations informed a four-part community engagement strategy that was executed by the CoVPN, supplementing local efforts undertaken by the clinical research sites. All Phase 3 COVID vaccine trials were IRB-approved by either the Advarra IRB, headquartered in Columbia, MD or Western IRB (WIRB) in Puyallup, WA.

### Part I: Utilization of community-based participatory research approaches to meaningfully involve communities throughout the research process

Increasing community awareness and knowledge to address and correct misperceptions, misinformation, and myths required the utilization of Community-Based Participatory Research (CBPR) approaches [21] and working with partners such as the National Institutes of Health (NIH) Community Engagement Alliance (CEAL) Against COVID-19 Disparities (https://covid19community.nih.gov/). Effective community engagement involves community members at all stages of the research. As protocols were being developed, a CoVPN CWG composed of clinical research site community engagement staff and CAB members was convened to offer insight into needed educational materials, review materials in development, and provide general guidance and direction for community engagement efforts. A CAB member and a Community Educator representative were also involved in reviewing each efficacy protocol and providing input into the informed consent materials. Ongoing capacity building and skills development ensured that community members had the tools and skills needed to meaningfully engage in protocol discussions.

In early conversations, our partners from the Indigenous Wellness Research Institute recommended the development and convening of priority population expert panels to discuss

each protocol in development, and generate reports detailing considerations and actions needed to ensure inclusion of BIPOC and older adult communities. Four US-based and one Latin American expert panel were convened (Native/Indigenous; Black/African American; Hispanic/Latino/a; Older Adult/Veterans). Each panel's members included 10–15 scientists and community leaders who identified with their respective priority population; represented diverse biomedical, social, and behavioral science expertise; and had dedicated their professional life to working with and within their communities. Ongoing communication with these panels also highlighted the need to address social determinants of health [22]. These discussions led to the early initiation of efforts to reduce participation burdens and costs for participants through the establishment of incentives that adequately reimbursed participants for their travel and other related study costs, as well as the acquisition and utilization of mobile units and satellite clinics, taking research to communities.

Relationships were vital in the success of our efforts to onboard clinical research sites at the four US Historically Black Medical Colleges and 3 clinical research sites specifically working with Tribal nations. Our guiding principal was to engage Tribal leaders and tribal communities, while acknowledging and respecting tribal sovereignty [23]. The NIH Tribal Health Research Office, the CoVPN Regulatory Affairs unit, and the pharmaceutical sponsors worked with Tribal nations to develop contracts that outlined tribal data, material, biospecimen sharing and ownership agreements. These contracts took time to negotiate and were not fully executed until the Moderna and Janssen trials were almost fully enrolled. The noticeable increase in AI/AN participants in the Novavax trial is likely due to these contracts being in place earlier during study accrual.

Involving community members and both elected and non-elected leaders in this process from the beginning ensured the use of respectful language (e.g., older adult vs. elderly, priority vs. target populations, American Indian/Alaska Native vs. Native Americans), inclusive identifiers (e.g., Asian AND Pacific Islander) and comprehensible materials (e.g., explaining safety pauses; expedited vaccine development processes; prevention of severe disease vs. prevention of acquisition). Access to these materials and information in English and Spanish was facilitated through the development of a US-focused website, the use of a toll-free call center with Spanish language capacity, and a publicly accessible Dropbox that included infographics, educational videos, social media posts, animations, and slide sets (TinyURL.com/CoVPN-Assets **Password:** CoVPNTria!$).

In addition to general public education, the materials drove COVID-19 vaccine inquiries to the website by referencing the URL (www.PreventCOVID.org). Each page on the website included a prominently displayed "Volunteer Now" link that directed interested parties to the Volunteer Screening Registry and its pre-screening survey. The survey collected contact information, demographics, and risk factors relating to employment and living conditions as predictors for risk of SARS-CoV-2 acquisition and development of severe COVID-19 illness. All US clinical research sites conducting COVID-19 vaccine studies had access to the pool of volunteers living in a set of pre-determined local zip codes. The registry database supported the enrollment of 30,000 or more people for each of the four CoVPN COVID-19 vaccine efficacy trials and allowed the clinical research sites to contact interested individuals about specific trials. Products included mRNA-based (Moderna), adenovirus-based (AstraZeneca [AZ] and Johnson & Johnson [J&J]), and subunit-based (Novavax) vaccines. Use of the Registry also allowed sites to focus their outreach efforts to particular demographics of interest as well as different risk factors as specified in any given clinical trial. As of April 2021, the registry has over 600,000 individuals who completed the survey. Efforts are currently underway to expand the registry to include: pediatric populations; gather data on SARS-CoV-2 infection and experiences with long-term COVID-19 disease; and to support future trials.

## Part II: Stakeholder engagement and building trust

Working in partnership with institutions and organizations with whom longstanding trusting relationships have been established is a vital component of community engagement, particularly in BIPOC communities who have a long history of and contemporary experiences with institutional racism and research ethics abuses. The HVTN and its clinical research sites have relied on such partnerships to assist in our efforts to communicate and share information with communities and potential participants with humility and authenticity. This involves knowing and understanding community context and needs, actively listening to fears and concerns of community members, being truthful and transparent at all times, ensuring that all information is provided in plain language, and making resources available from trusted sources to help community members make informed decisions. COVID-19 quickly became politicized, increasing perceptions of systems and research institutions as untrustworthy and creating fear and uncertainty about the vaccine clinical trials and vaccination in general. Partnerships were instrumental in developing and implementing activities that utilized trusted voices to address questions and concerns about safety and side effects, equitable inclusion in vaccine trials, and the pace of vaccine development [24]. These partnerships included social service providers, advocacy organizations, physician and medical professional associations, media, academic institutions, local/state/national government partners, and faith-based organizations, particularly those who serve BIPOC communities. Outreach to essential worker organizations and corporations, such as meat packing plants, nursing homes and assisted living facilities, and agricultural industries was critical to these efforts. Forming these new partnerships opened important channels of communication and information dissemination. Building and enhancing trust also involved utilizing our partnerships with leaders of these organizations as "trusted voices," persons to whom community members could ask questions and obtain the information needed to make informed health and medical decisions. Our efforts in this area also supported bringing many of these organizations and community leaders together to co-coordinate and co-host COVID-19 educational sessions for their communities, providing another opportunity for communities to see unified trusted voices sharing science and addressing community concerns regarding COVID-19.

## Part III: A faith initiative

The CoVPN Faith Initiative leveraged the breadth of established relationships from the HVTN's history of successful engagement in faith communities. Through these efforts, a faith-based advisory council was established to provide guidance and direction for community engagement efforts with faith-based groups, and to implement a national faith-focused CoVPN education program that used anti-racism, anti-homophobic, anti-transphobic, and other principles to ensure that the activities and messages reached broad audiences. Six Faith Ambassadors, representing clergy from a variety of faith traditions, were identified across the US to support educational activities and speak to the intersection between faith and science, with a focus on BIPOC communities. Faith Ambassadors worked closely with more than 30 regional faith leaders to engage congregations and faith-based organizations and to identify additional channels for message dissemination.

## Part IV: Communications and community influencers

Following the launch of the CoVPN website and Registry in July 2020, an extensive marketing and communications campaign launched in September 2020 to address COVID-19 vaccine trial concerns. The campaign focused on adults over 50 years old and Latino/a/Hispanic and Black/African American communities. It was developed using audience insights and testing

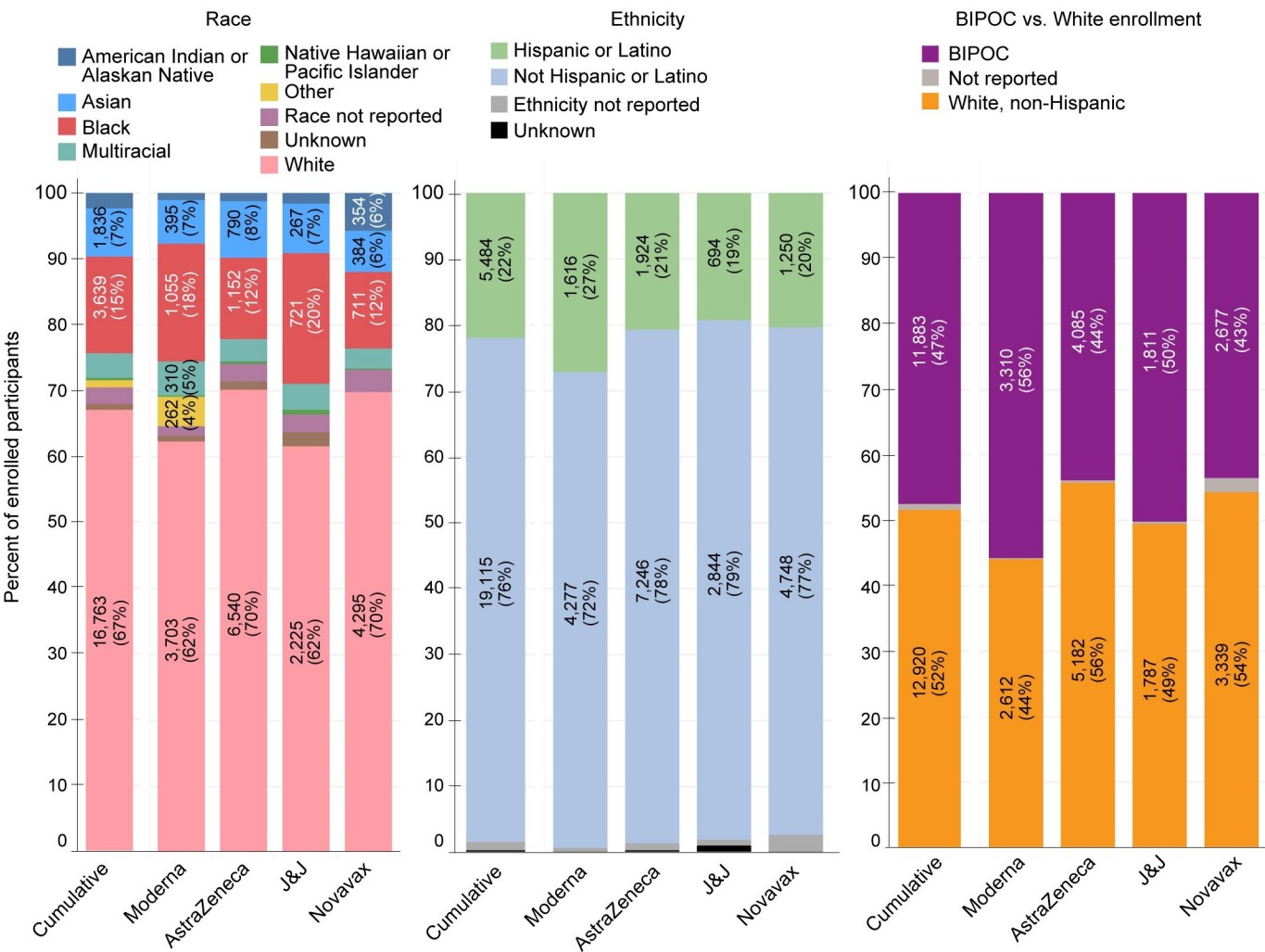

**Fig 1. US diverse enrollment across the NIH-funded Moderna, AstraZeneca, J&J, and Novavax trials.**

gained through in-depth one-on-one interviews and surveys conducted in English and Spanish with members of the priority populations. The campaign employed a robust media mix including TV, connected TV, radio, internet audio, digital platforms and social media, as well as partnerships and sponsorships with trusted organizations such as the American Association of Retired People, BlackDoctor.org and celebrity personalities. Under the umbrella theme, "Help End the Uncertainty," the campaign consisted of hundreds of content pieces in Spanish and English–broadcast and digital ads, sponsored content, videos, quizzes, interviews, testimonials–and advertising spots that combined user-generated testimonials with a Harrison Ford voice-over. The campaign achieved over 500 million gross impressions, resulting in over 5 million website visits. The advertising buy focused on Black and Latino/a adults, aged 45 and older, who lived in the US and speak either English or Spanish.

Public requests for greater transparency about the clinical trials, calls for explanations of the vaccine science in lay language, and the need for the voices of leading scientists to speak more directly to communities drove the development of content for the communications platform. To respond to these requests, the CoVPN established a blog, "COVID-19 Vaccine Matters," initially housed on the Johns Hopkins Coronavirus Resource website (https://coronavirus.jhu.edu), and

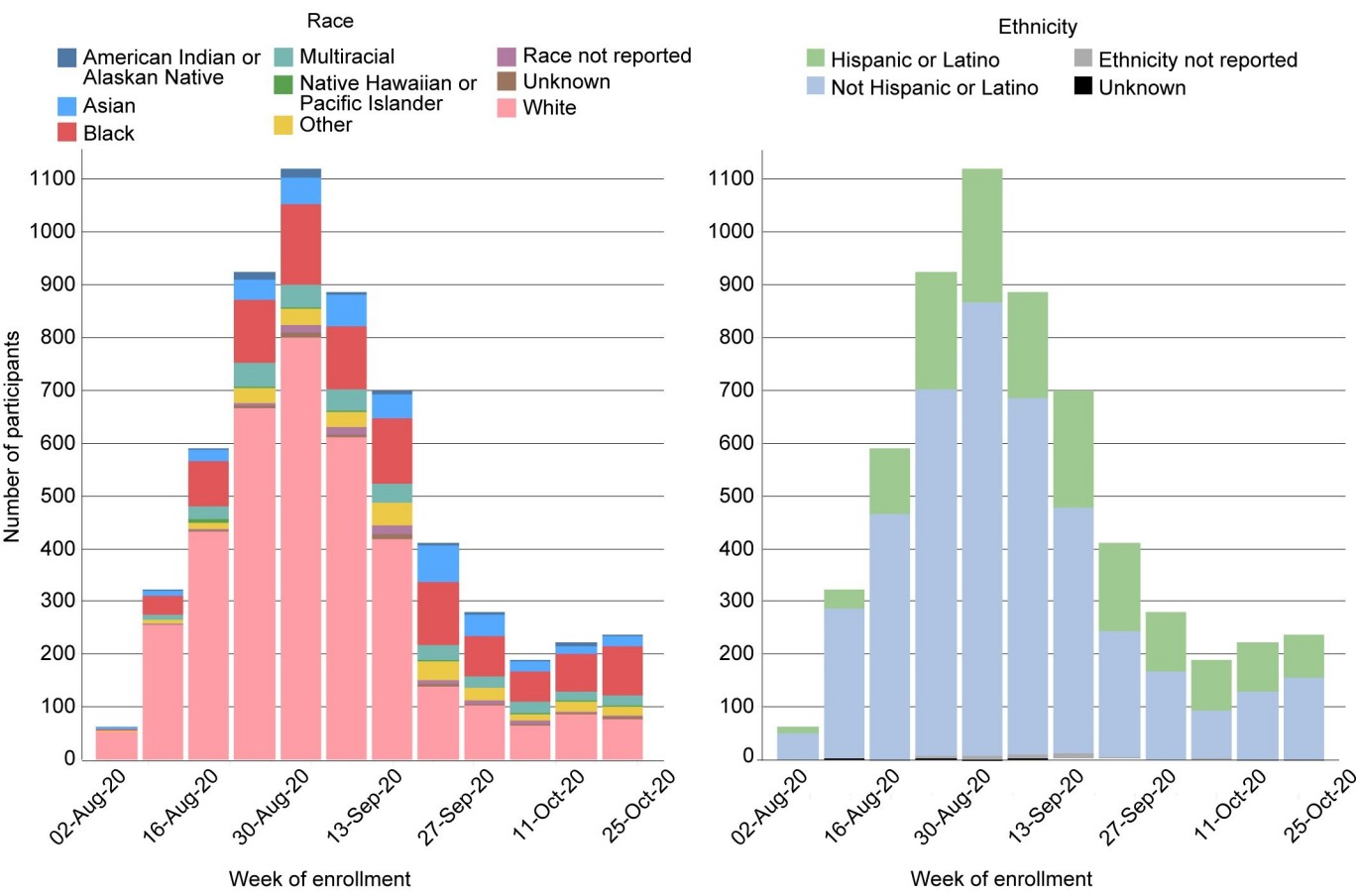

**Fig 2. Moderna (Cove study) enrollment.** (A) Counts reflect randomized participants.

later hosted jointly with the University of Washington. The blog was launched in November 2020 and has steadily grown in readership to a current 22,000 readers of each blog post (March 2021). The blog provided clear, current, and engaging information on the trials as they progressed, addressing issues of concern and giving stakeholders a "front-row seat" as the science unfolded.

To assess the effectiveness of the community engagement strategy, we analyzed the racial and ethnic demographic data of enrolled participants across the Moderna, AZ, J&J, and Novavax trials for the US-based CoVPN-affiliated clinical research sites only. The full data reflecting other independently contracted clinical research sites are not available for all study sponsors. Across the trials, enrollment was defined differently depending on the trial sponsor: as participant randomization and/or completion of Study Day 1 with intention of continuing (Moderna, AZ, J&J), or completion of Study Visit 2 (Novavax). Racial and ethnic category data were collected based on the established NIH-required racial and ethnic categories and definitions [25]. Race focuses on physical characteristics, particularly skin color, whereas ethnicity attempts to capture a group's cultural identity.

## Results

Across the four clinical trials for which data are available, 47% of participants enrolled at CoVPN sites in the US were BIPOC (Fig 1). This included 2% American Indian/Alaska Native,

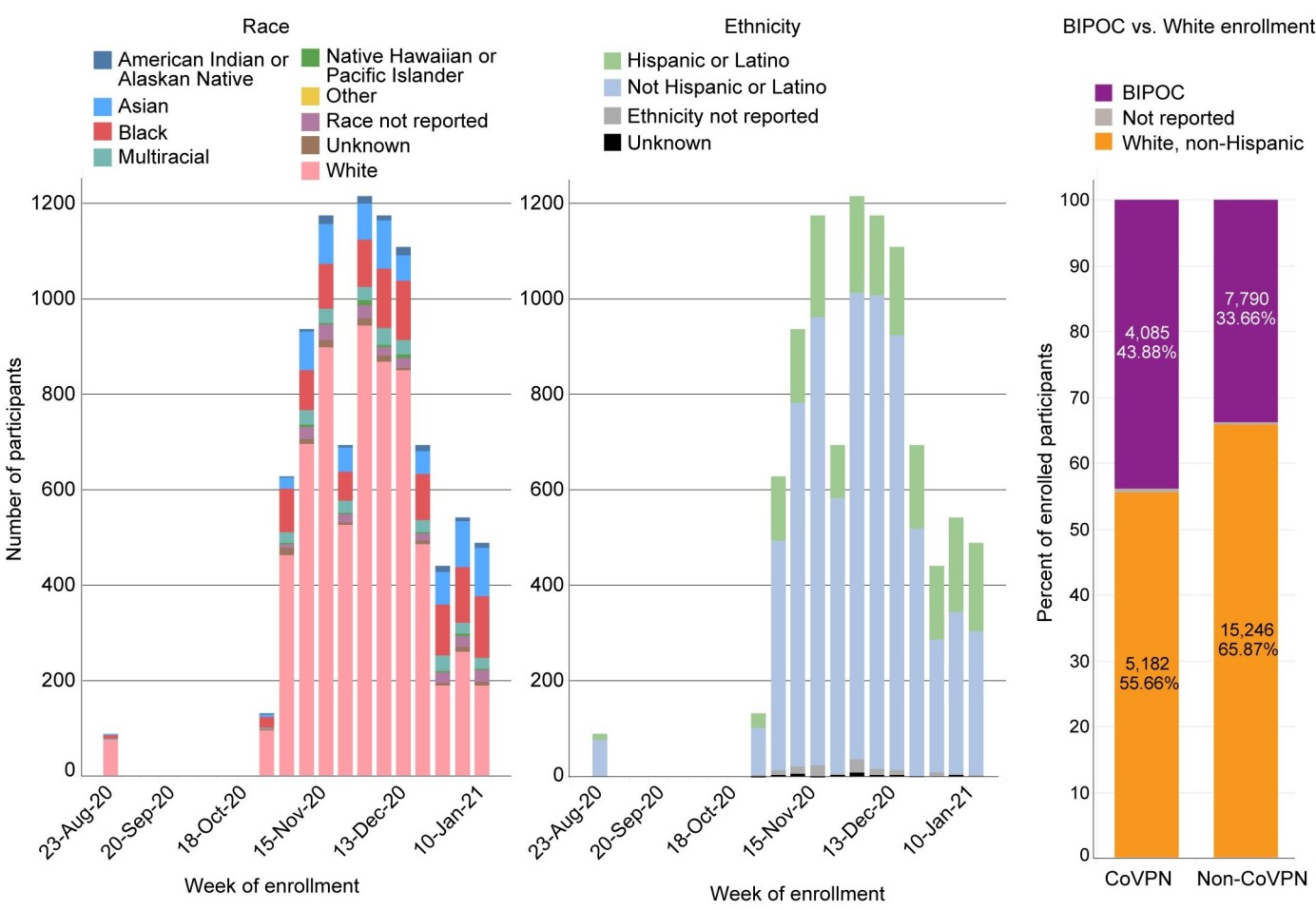

**Fig 3. AstraZeneca study enrollment.** (A) Counts reflect randomized participants.

15% Black/African American, 0.36% Hawaiian/Pacific Islander and 7% Asian. A total of 5,485 (22%) identified as Hispanic/Latino/a. Across the trials, enrollment of White participants ranged from 44% (Moderna) to 56% (AZ), and the enrollment of BIPOC communities closely mirrored their composition in the larger US population. Across all trials and racial/ethnic groups, enrollment at week 1 was low.

Data for just the CoVPN sites are available for the Moderna trial, the first trial to begin and complete enrollment. These data show a relatively lower rate of enrollment across all BIPOC groups, compared to the enrollment of White non-Hispanic participants, in the first week of enrollment (Fig 2). Within two weeks, White enrollment began to quickly outpace enrollment of BIPOC participants, and this continued throughout the accrual period. Although BIPOC enrollment increased over time, it never approached the rate of White enrollment. This reality required actions to be taken to ensure that there were enough allocated enrollment slots remaining to be filled by BIPOC individuals. As a result, all clinical research sites were instructed to first slow (September 11, 2020), and then halt (September 30, 2020) White enrollment. These actions allowed the remaining enrollment slots to be filled by BIPOC individuals, thus ensuring that eventual safety and efficacy data would be relevant to the US populations where needed most.

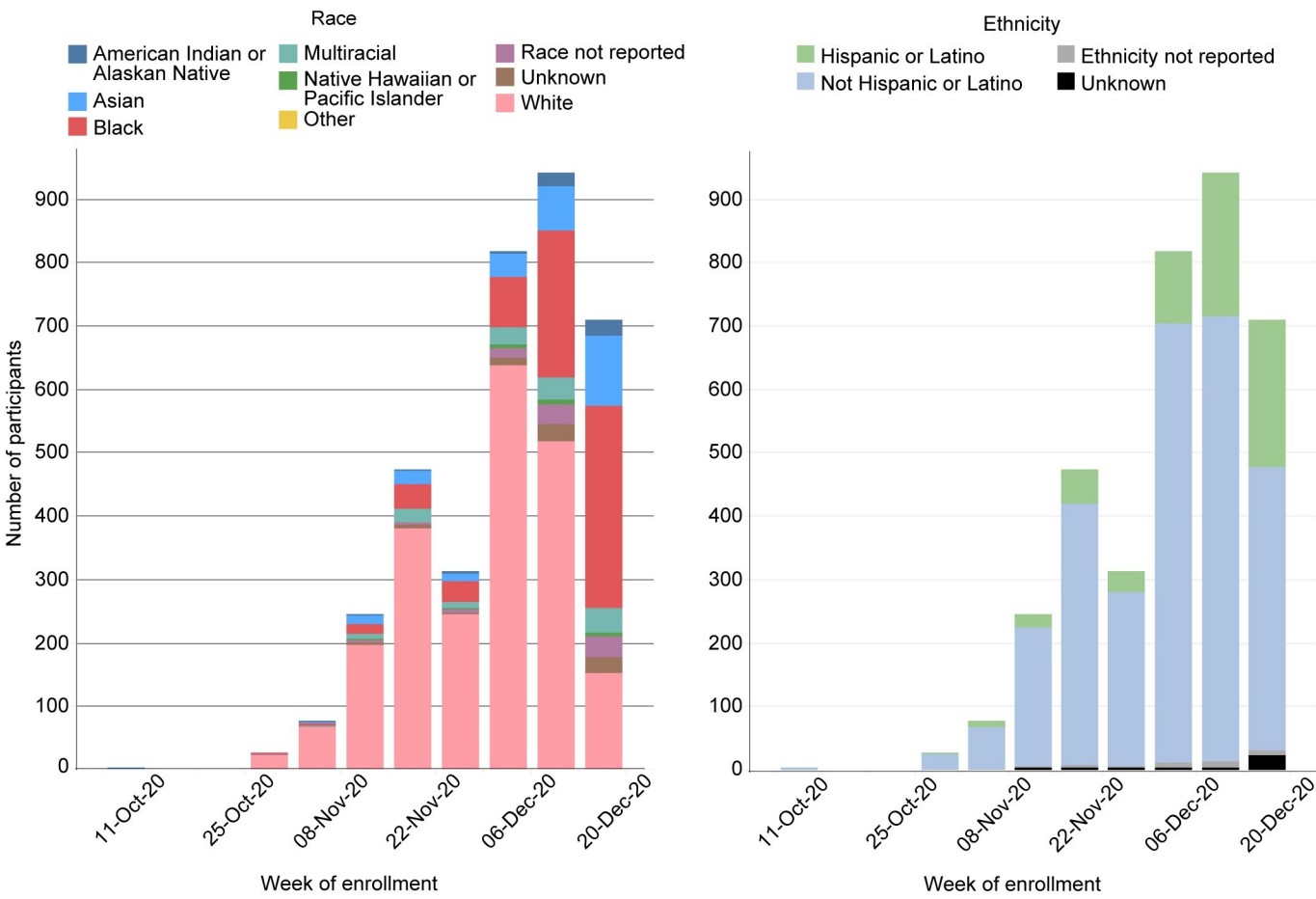

**Fig 4. J&J (ensemble study) enrollment.** (A) Counts reflect participants who completed study Day 1 with the intention of continuing.

In the AZ (Fig 3) and J&J (Fig 4) trials, White enrollment again outpaced that of BIPOC participants. Enrollment across all BIPOC groups was low during the first month of enrollment and after a study safety pause in both trials, enrollment was slow to resume for two more weeks, followed by a sharp uptick in enrollment. As with the Moderna trial, BIPOC enrollment began slowly but was steady and always outpaced by White enrollment. Focused and intentional efforts to enroll BIPOC individuals were accelerated in the final weeks of the trials to ensure that remaining enrollment slots were filled by BIPOC individuals.

The Novavax trial, which opened five months after the Moderna trial, benefitted considerably from ongoing community engagement efforts, and greater and more diverse volunteer registry records. This was particularly true of efforts to partner with tribal leaders to address data sovereignty and ownership, resulting in increased participation among Indigenous peoples. As a result, although enrollment of BIPOC individuals was always outpaced by that of White individuals, it was constant throughout the Novavax accrual period (Fig 5). In general, the CoVPN sites were more successful in recruiting BIPOC participants than non-CoVPN sites (Figs 3 and 5).

## Discussion

Intentional and robust community engagement efforts are critical to ensuring the equitable inclusion of BIPOC communities in vaccine clinical trials. Equitable inclusion requires

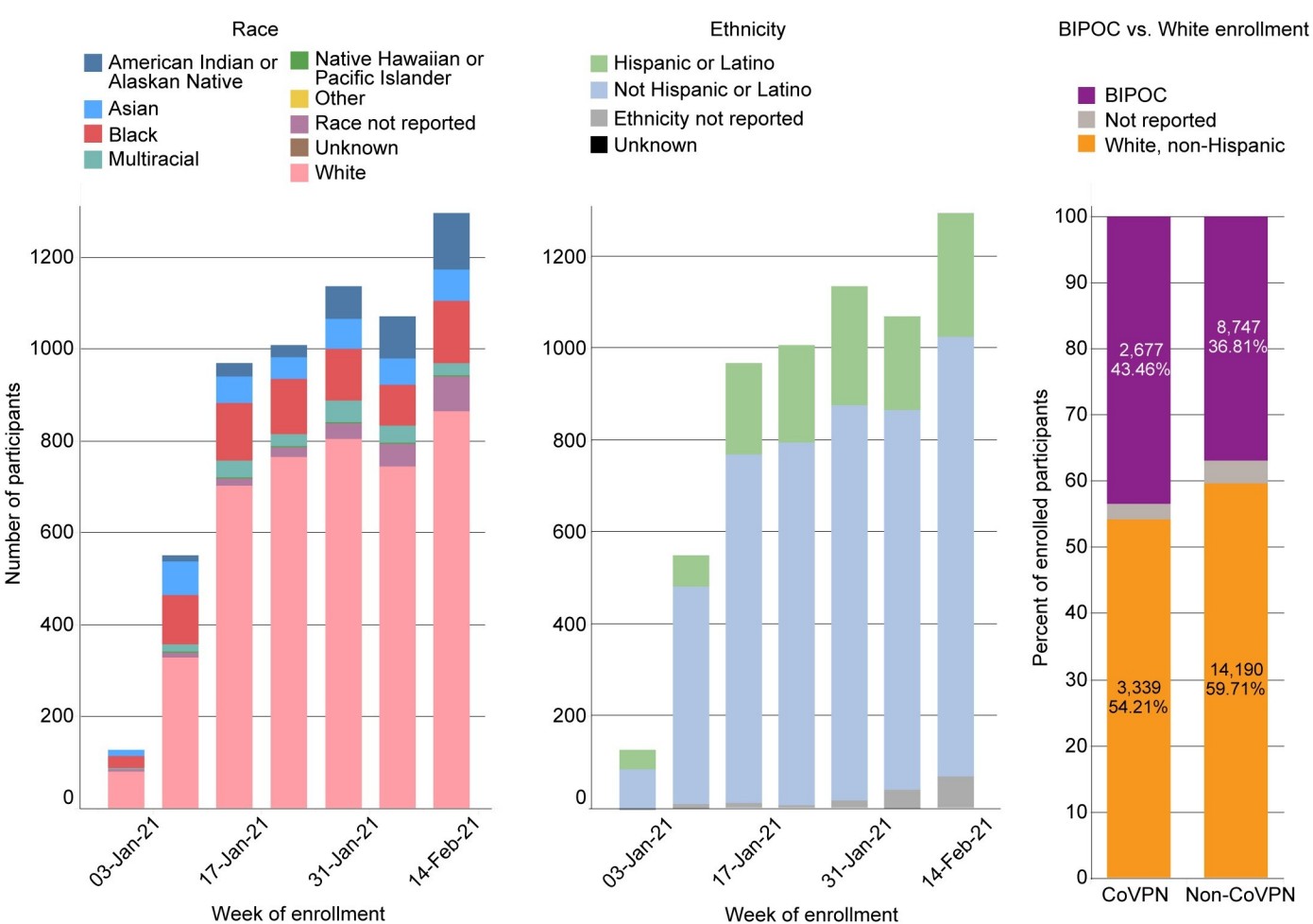

**Fig 5. Novavax study enrollment.** (A) Counts reflect participants who completed study visit 2 or 2.1.

representation of BIPOC participants reflecting the composition of the US population in situations where disease impact is equally distributed across communities. However, an approach that enrolls BIPOC participants at a rate that reflects the disproportionate impact of the disease on specific populations would represent an optimized framework for clinical trial enrollment objectives. If the intended use of a biomedical intervention is more impactful in certain communities, then the clinical trial enrollment should be informed by the eventual intended use of such biomedical interventions.

These data illustrate that with sufficient resources, commitment and community engagement expertise, the equitable enrollment of BIPOC individuals can be achieved. What is also clear, however, is that even with robust fiscal resources and a longstanding collaborative and collective effort, the enrollment of White persons outpaces that of BIPOC communities. Without established recruitment goals that reflect the slower yet steady pace of BIPOC enrollment, the allocated enrollment slots were quickly filled, effectively blocking BIPOC persons' opportunities for participation. Rather than directing sites to slow or halt White enrollment, which presents its own operational challenges, future vaccine clinical trial efforts must include clear established goals for BIPOC enrollment from the outset of study accrual, reserving space in the trial to ensure equitable inclusion. The establishment of recruitment goals has achieved remarkable success in recruiting BIPOC and transgender participants in HIV Prevention

Trials Network (HPTN) 083, an HIV prevention trial that set specific and measurable goals for the enrollment of transgender women and Black cisgender men who have sex with men [26].

Another approach to ensuring equity is the development of population-specific trials. HPTN 091 will be the first HPTN study designed specifically and exclusively for transgender women. AIDS Clinical Trials Group (ACTG) A5366 is the first HIV cure-related study designed specifically and exclusively for cisgender women. The ACTG is also currently developing a study exploring the effect of gender-affirming hormones on the HIV reservoir in transgender women living with HIV. These population-specific approaches ensure the inclusion of under-represented populations in research that could benefit them.

When conducting clinical trials, research teams can utilize the framework proposed by Bolen et al (2006) to select consistent a priori recruitment goals for underrepresented groups based on the research question and study location [27]. The Division of AIDS (DAIDS)-funded Office of HIV/AIDS Network Coordination (HANC) Legacy Project recently developed the Representative Studies Rubric tool to ensure representation of priority populations in the development of research protocols [28], and advocacy efforts are underway to ensure the adoption of this rubric for DAIDS-funded research.

As seen in the Novavax trial, it is clear that prolonged and directed engagement with priority communities can yield equitable inclusion. Ongoing commitment to these standards and partnerships will help communities view researchers and research institutions as trustworthy, and build and strengthen rapport between communities, researchers, and research institutions. To this end, the recent publication by the US Food and Drug Administration of guidelines for the pharmaceutical industry to emphasize diversity of enrollment in clinical trials is a welcome step [29].

## Acknowledgments

We thank the over 130,000 individuals who volunteered in these trials. We are grateful to Jasmine Hwang and Rohit Banerjee from SCHARP for their assistance in the creation of the datasets and Mindy Miner for her editing and figure development assistance. We thank the CoVPN Expert Panelists and Community Working Group members for their unwavering commitment to their communities and their guidance and direction. We also thank the CoVPN Community Engagement team for its collaborative efforts and commitment to equity and inclusion.

**Disclaimer:** The views expressed in this article are those of the authors and should not be construed to represent the official positions of their affiliated institutions, the US Army, the Department of Defense, or the National Institutes of Health.

## Author Contributions

**Conceptualization:** Michele P. Andrasik, George A. Mensah.

**Data curation:** Richa Chaturvedi.

**Formal analysis:** Michele P. Andrasik, Richa Chaturvedi.

**Methodology:** Michele P. Andrasik, Gail B. Broder, Stephaun E. Wallace, Nelson L. Michael, Chris Beyrer, David R. Wilson, James G. Kublin, George A. Mensah.

**Project administration:** Michele P. Andrasik, Nelson L. Michael, Jasmin Aina.

**Supervision:** Michele P. Andrasik, Nelson L. Michael, James G. Kublin.

**Writing – original draft:** Michele P. Andrasik.

**Writing – review & editing:** Michele P. Andrasik, Gail B. Broder, Stephaun E. Wallace, Richa Chaturvedi, Nelson L. Michael, Sally Bock, Chris Beyrer, Linda Oseso, Jonathan Lucas, David R. Wilson, James G. Kublin, George A. Mensah.

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
