## [Decision Letter · Decision Letter 0]

1 Jul 2021

PONE-D-21-14138

Increasing Black, Indigenous and People of Color Participation in Clinical Trials Through Community Engagement and Recruitment Goal Establishment

PLOS ONE

Dear Dr. Andrasik,

Thank you for submitting your manuscript to PLOS ONE. After careful consideration, we feel that it has merit but does not fully meet PLOS ONE’s publication criteria as it currently stands. Therefore, we invite you to submit a revised version of the manuscript that addresses the points raised during the review process.

Both reviewers thought the manuscript highly important and valuable. They each requested additional details in the background; greater specificity in the methods section, in particular; and a few clarifications in the results. In accordance with Reviewer 2’s suggestion to reduce the usage of less common acronyms (in addition to further comments on the attached PDF), this would improve readability of the manuscript.

We look forward to receiving your revised manuscript.

Kind regards,

Peter A Newman, Ph.D

Academic Editor

PLOS ONE

Journal Requirements:

Additional Editor Comments (if provided):

Reviewers' comments:

Reviewer's Responses to Questions

**Comments to the Author**

1. Is the manuscript technically sound, and do the data support the conclusions?

Reviewer #1: Yes

Reviewer #2: Yes

2. Has the statistical analysis been performed appropriately and rigorously? 

Reviewer #1: N/A

Reviewer #2: Yes

3. Have the authors made all data underlying the findings in their manuscript fully available?

Reviewer #1: Yes

Reviewer #2: Yes

4. Is the manuscript presented in an intelligible fashion and written in standard English?

Reviewer #1: Yes

Reviewer #2: Yes

5. Review Comments to the Author

Reviewer #1: Thank you for the opportunity to review this manuscript, which addresses an important topic related to representation of BIPOC individuals in clinical trials, which has implications for equitable dissemination and implementation based on clinical trial findings. The comments below are intended to further strengthen this manuscript:

Introduction

- The authors mention the “Good Participatory Practice” guidelines; could these be described briefly for a general readership that may not be familiar with these HIV specific guidelines?

Methods

- Within “Part 1” (particularly the first paragraph) – it would be helpful to see more specificity in terms of what the CBPR efforts looked like in terms of community providing feedback (vs. just the outcomes of the feedback that they provided). This might help other researchers to better emulate the authors’ efforts.

- Pg. 4, line 145, “the registry has over 600,000 diverse individuals…” Would be helpful to define what they mean by “diverse” (e.g., race/ethnicity? Sexual orientation? Age?)

- “Part II – Involving Communities” – could the authors clarify how this is different than Part I (CBPR), which also is focused on community involvement? For example, Part II describes building relationships with groups through Historically Black Medical Colleges and engaging tribal/indigenous communities, which sounds closely aligned with a CBPR approach.

- “Part III – stakeholder engagement” – it may help readers to replicate the authors’ efforts to describe what interpersonal skills/behaviors they used in order to demonstrate “humility and authenticity” to participants.

- “Part V” mentions in-depth one-on-one interviews and surveys to shape the marketing and communications campaign. If the findings from these interviews and surveys have been published somewhere (even if not in an academic journal), it would be great to reference them. The same paragraph mentioned reaching over 500 million gross impressions and 5 million website visits – do the authors have any data on who (e.g., demographically) was reached?

- Relatedly, the authors mention developing a blog to reach lay audiences regarding the vaccine. Do they have any data on who the readership (e.g., demographically) of the blog was?

Results

- It would be helpful to clarify what the ideal enrollment would have been for these trials in terms of enrolling BIPOC participants. For example, do the authors believe that ideally, representation of different groups should reflect the US population? The proportion of the population affected by COVID-19? This comes up in the discussion too, when the authors refer to “the equitable enrollment of BIPOC individuals.” How do the authors define “equitable enrollment” (e.g., equitable in terms of representation reflecting the impact of COVID by demographic group?)?

Discussion

- The authors provide important recommendations to help with planning future clinical trials, such as setting “clear established goals for BIPOC enrollment…” Do the authors have recommendations, based on their experience, on what these goals should be (or guidelines on how to determine what these goals should be) in order to achieve equitable representation?

Minor

- Would be great to reduce use of lesser known acronyms (e.g., CRS for clinical research sites)

Reviewer #2: The article is extremely important, timely, and well-written. Below, I provide a few suggestions for improvement.

General

Some proofreading is needed. I made several suggestions in the attached, but a careful review would help.

Background.

The background would benefit from some data on underrepresentation of BIPOC populations in clinical trials.

Methods

A brief discussion of how data on race and ethnicity were collected in the trials and presented in the paper would be of help. It appears they were collected separately and per OMB and that people were able to select multiple races, but audiences/readers are consistently confused by these data, particularly when race and ethnicity are presented separately as is done here.

Sentence on higher enrollment of BIPOC in CoVPN sites belongs in the results.

Results

I provided a few suggestions in the tracked-changes document to increase clarity of how findings are described.

May want to point out that Week 1 enrollment was low across all trials and groups.

Addressing data sovereignty and ownership is mentioned in 2-3 places in the paper, would be helpful to specify how it was addressed or at least provide an example from one trial.

Discussion

Well written and argued. I only struggled with this sentence because I think it is an overstatement: "When this is the reality across clinical research, the establishment of recruitment goals and population-specific trials may no longer

be necessary, as equitable inclusion will be the norm and not the exception." Given that disparities in healthcare treatment and access would persist even in this case, I suggest the authors instead point out that ongoing commitment, to these standards and partnerships will decrease the cost involved in community engagement for any one study.

Figures:

-I am unclear where Alaska Natives and Pacific Islanders are in these figures, if at all.

-Why is it that there are data provided for people with unknown ethnicity, but not unknown race, especially given that Latinos often select unknown or do not specify race?

-On the left panel of the 1st & 5th figure, orange is for Asians. However, on the right panel orange is for non-Hisp/Whites. I suggest using a different color for one.

6. PLOS authors have the option to publish the peer review history of their article (what does this mean?). If published, this will include your full peer review and any attached files.

Reviewer #1: No

Reviewer #2: **Yes: **Nina Harawa

---

## [Author Response · Author response to Decision Letter 0]

3 Aug 2021

This is attached in the response to reviewers document attached

---

## [Decision Letter · Decision Letter 1]

7 Oct 2021

Increasing Black, Indigenous and People of Color Participation in Clinical Trials Through Community Engagement and Recruitment Goal Establishment

PONE-D-21-14138R1

Dear Dr. Andrasik,

We’re pleased to inform you that your manuscript has been judged scientifically suitable for publication and will be formally accepted for publication once it meets all outstanding technical requirements.

Kind regards,

Peter A Newman, Ph.D

Academic Editor

PLOS ONE

Additional Editor Comments (optional):

Reviewers' comments:

Reviewer's Responses to Questions

**Comments to the Author**

1. If the authors have adequately addressed your comments raised in a previous round of review and you feel that this manuscript is now acceptable for publication, you may indicate that here to bypass the “Comments to the Author” section, enter your conflict of interest statement in the “Confidential to Editor” section, and submit your "Accept" recommendation.

Reviewer #1: All comments have been addressed

2. Is the manuscript technically sound, and do the data support the conclusions?

Reviewer #1: Yes

3. Has the statistical analysis been performed appropriately and rigorously? 

Reviewer #1: N/A

4. Have the authors made all data underlying the findings in their manuscript fully available?

Reviewer #1: Yes

5. Is the manuscript presented in an intelligible fashion and written in standard English?

Reviewer #1: Yes

6. Review Comments to the Author

Reviewer #1: (No Response)

7. PLOS authors have the option to publish the peer review history of their article (what does this mean?). If published, this will include your full peer review and any attached files.

Reviewer #1: No

---

## [Editor Report · Acceptance letter]

11 Oct 2021

PONE-D-21-14138R1 

Increasing Black, Indigenous and People of Color participation in clinical trials through community engagement and recruitment goal establishment 

Dear Dr. Andrasik:

I'm pleased to inform you that your manuscript has been deemed suitable for publication in PLOS ONE. Congratulations! Your manuscript is now with our production department. 

Kind regards, 

on behalf of

Dr. Peter A Newman 

Academic Editor

PLOS ONE